# Evolution in the Impact of Pro-Poor Policies on Farmers' Confidence: Based on Age-Period-Cohort Analysis Perspective

Zheng Wang [1], Mingwei Yang [2,*], Kailu Guo [1,*], Zhiyong Zhang [3] and Ying Shi [4]

[1] School of Economics and Management, Taiyuan Normal University, Jinzhong 030619, China; wz@tynu.edu.cn
[2] Library, Qinghai University, Xining 810016, China
[3] Institute of Nationalities, Guizhou Academy of Social Sciences, Guiyang 550002, China; 2020651920@email.ctbu.edu.cn
[4] School of Computer Science and Technology, Taiyuan Normal University, Jinzhong 030619, China; sy@tynu.edu.cn
[*] Correspondence: 2020651909@emall.ctbu.edu.cn (M.Y.); guokl@tynu.edu.cn (K.G.)

**Abstract:** The Age-Period-Cohort Model is used in this paper to examine how farmers' confidence has changed in response to various measures for reducing poverty, based on data from 13,559 household tracking surveys, with a view to inform rural poverty reduction policies within Targeted Poverty Reduction Strategy (TPRS). The findings indicate that: (1) Farmers who get monetary grants have significantly lower levels of confidence than farmers who do not. The difference between the ages of 18 and 70, where this issue is more noticeable, grew between 2013 and 2018. (2) Between 2010 and 2018, transfer employment was more likely than monetary handouts to increase farmers' confidence, and this difference was particularly obvious among young people (18–45 years old) and elderly individuals (65+). (3) The confidence gap between farmers with and without medical insurance has widened over time. Farmers with medical insurance have significantly higher confidence than farmers without it. Lessons for TPRS suggest that to reduce poverty among poor groups in a way that is both stable and sustainable, poverty alleviation strategies should take psychological factors into account when evaluating their efficacy. They should also concentrate on how employment boosts self-confidence.

**Keywords:** pro-poor policies; confidence; age-period-cohort analysis; farmers

## 1. Introduction

Anti-poverty theory and practice focus on issues related to the subjectivity and sustainability of the pro-poor target. Whether the government poverty alleviation policy can inspire farmers' confidence in pursuing a better life, or to rely on the psychology of dependence and make the policy implementation [1–3]. Low income is just a manifestation of poverty. In the long run, the lack of endogenous motivation for individuals to yearn for a better life, resulting in negative idleness is the key factor in poverty traps [4]. The theory is fully manifested in the targeted Poverty Reduction Strategy implemented in China.

Poverty is not only an economic burden but also a psychological one, and stigma and poverty stigma can lead to confidence failure among the poor [5]. Material poverty and spiritual poverty are the two main manifestations of the human poverty problem [6–8]. For the poor, it is a high-pressure scarcity environment in which they have to live all the time. Growing up in such an environment creates a scarcity mentality in the poor, which in turn makes them prone to limit themselves to their immediate interests and neglect their long-term plans. In the long run, this leads to a vicious cycle of "poverty-scarcity mentality and behavioral patterns-continued poverty" [9].

Spiritual poverty, represented by a lack of confidence and confidence failure, is often the essential characteristic of materially poor groups in general [10,11]. Compared with material poverty, spiritual poverty has the distinctive characteristics of being difficult to

quantify, long-term, and hidden [12,13]. In terms of the interrelationship between the two, spiritual poverty and material poverty often go hand in hand, and spiritual poverty is both a result of material poverty and an important cause of the persistence of material poverty [14]. Some scholars point to dysfunctional confidence as a cause of poverty, with low levels of confidence leading to low levels of future investment [15–17]. However, some scholars argue that confidence failure is a consequence of poverty, that poverty exacerbates the negative effects of behavioral bias on confidence choice among the poor, and that an individual's economic status affects his or her level of confidence [18–20]. Poverty is more than simple material deprivation [21]. Due to the threat of stereotypical images, people can lower their expectations and behavioral expectations when they are aware of their poor and vulnerable status [22]. Psychological research has also shown that decision-making under constraints or difficult trade-offs can deplete individual confidence [23].

Why do the poor have relatively low levels of ambition? The social cognitive theory of social stratification explains that due to chronic material poverty, the poor realize that the achievement of their goals depends heavily on external factors, and over time develop a social cognitive tendency toward contextualism [24–27]. The social cognitive orientation leads the poor to exhibit psychological and behavioral characteristics that are significantly different from those of other groups in terms of social interaction and self-perception [28,29]. Most of these characteristics are detrimental to the social adjustment of the poor. For example, they attribute the gap between rich and poor to external factors such as birth and luck, and believe that their efforts can make little difference, so they lose confidence, settle for the status quo and lose all motivation to escape poverty [30–32].

What kind of poverty alleviation policies can effectively stop the vicious cycle of poverty? In their studies of contextualist social cognitive dispositions, scholars have proposed a range of pro-poor programs designed to raise the level of aspirations of the poor. For example, the Conditional Cash Transfers program, a popular long-term multidimensional poverty reduction program in developing countries, is centered on "using cash incentives to stimulate the use of public services (mainly education and health) by low-income groups. In Mexico, the program has increased the motivation of parents from poor families to finance their children's college education by 11%, while children's expectations to attend college themselves increased by 20% [33]. Poverty alleviation policy tools include both cash support and livelihood skills assistance [34]. Hoang et al. found that adding a household member involved in non-farm activities reduced the probability of poverty by 7–12% and increased household expenditures by 14% over two years [35]. Current studies have mainly focused on the evaluation of the effects of a particular poverty alleviation policy, with few comparative analyses of the effects of different policies. In this paper, we will compare and analyze the evolution of the effects of three types of poverty alleviation policies: Cash grants, transfer employment, and medical insurance.

Since 2013, the Chinese government has formally implemented Targeted Poverty Reduction Strategy, whose goal is to reduce poverty by more than 10 million people per year [36,37]. By 2020, the incidence of rural poverty in China had fallen from 10.2% to 0.6%, basically eliminating absolute poverty among the rural population [38,39]. However, the experience of China's Targeted Poverty Reduction Strategy shows that while material poverty among rural residents is decreasing, spiritual poverty, such as poor people's confidence failure with the status quo, is repeatedly highlighted [40,41]. "First aid the poor and support the confidence" has become an important consensus in the Chinese government's poverty alleviation work.

To date, the impact of age-cohort and period interactions on farmers' poverty reduction strategies has not been reported, and the trend by which the farmers' confidence is influenced in poverty reduction evolution is still unclear. This article will focus on the above issues, and the main purposes are to:

(1)   Explore the dynamics between Targeted Poverty Reduction Strategy and farmers' confidence, with APC analysis to isolate changes in the correlation between the two at individual age, period, and cohort.

(2) Compare the difference of inhibition effect of three specific precise poverty alleviation policies, namely cash grants, transfer employment, and medical insurance, on farmers' confidence.

(3) Analyzing the changing trends in farmers' confidence in poverty from a life-cycle and life-course perspective.

## 2. Materials and Methods

### 2.1. Methods

Age-Period-Cohort Model (APC) was first proposed by Frost in 1939 to conduct a long-term trend of tuberculosis mortality [42]. APC is demographic concept that must be considered in the long-term trend analysis of poverty-related variables of the population [43–46]. Previous studies of trends in poverty have tended to focus on one dimension, for example, that poverty changes over time or that poverty varies by age [47–50]. However, these studies have overlooked an important factor, namely that the interaction of age, cohort, and period may influence the observed trends. Obtaining realistic age, period, and cohort effects is important to understand the mechanisms of poverty formation and the factors. Age effects usually refer to the physical, psychological, and social changes associated with changes in biological age [51,52]. In this paper, the age effect reflects age-led changes in physical functioning as well as changes in social status. The age effect reflects the impact of intrinsic forces such as age-led changes in physical functioning and changes in social roles on farmers' confidence [53]. Period effects represent changes associated with changes in periods, which are usually caused by the transient effects of external environmental factors at a given time, including special historical events, changes in the socioeconomic environment, and new technological breakthroughs [54,55]. The most striking feature of period effects is that their impact is uniform across all age groups at the same time. The cohort effect is reflected in the fact that its effects are concentrated on groups of individuals born at the same time. Some historical events or social changes that these individuals experience together gradually produce the same imprint on them, influencing peasant confidence through slow accumulation or delayed occurrence by the interaction of external forces (e.g., major social events) and internal forces (e.g., individual development processes) [56–58]. Therefore, we need to use the lens of social change to explain the utility of these macro factors in the poverty reduction efforts of Chinese farmers.

Age, period, and cohort are all time-dependent variables and the relationship between the three can be expressed as: period = age + birth cohort, which inevitably leads to the problem of perfect covariance between the three [59–62]. The problem of how to deal with the complete covariance between age, period, and cohort and to solve the problem of parameter estimation has always been a problem. The problem of parameter estimation has always troubled researchers. They did not address the issue of age, period, and cohort covariance [63,64]. Subsequent researchers have explored this area in depth, launching a series of studies that have developed different research paths such as two-factor models, non-linear parameter transformations, proxy variable methods, IE variables, etc. [65–68]. Based on previous studies, Yang et al. proposed the Hierarchical APC (HAPC) model, which is also known as the Hierarchical APC-Cross-Classified Random Effects Models (HAPC-CCREM) for multi-period cross-sectional survey data [55]. HAPC-CCREM avoids the problem of complete covariance between the three by placing the age effect and the period and cohort effects at different levels of the model, which solves the problem of model identification while enabling differences in period and cohort to be observed [57]. The underlying assumption is that age is an individual-level variable, whereas people in the same period or cohort experience similar social events and life experiences and will have similar effects at the group level, and both period and cohort can be considered group-level variables [55]. The HAPC-CCREM models are expressed as follows.

Individual level (Level 1):

$$Y_{ijk} = \beta_1 AGE_{ijk} + \beta_2 AGE_{ijk}^2 + \beta_3 X_{ijk} + \beta_4 Z_{ijk} + \varepsilon_{ijk} \tag{1}$$

where $Y_{ijk}$ represents the level of confidence of respondent $i$ who belonged to birth cohort $j$ in survey period $k$; $\beta$ represents the regression coefficient, $Z_{ijk}$ represents the group of poverty alleviation policy variables (including industrial cash grants, shift employment, and medical insurance). $X_{ijk}$ represents the group of control variables at the individual level (including gender, ethnicity, education, health, marital status, socio-economic status, and regional category).

Group level (Level 2):

$$\beta_{0jk} = \gamma_0 + \mu_{0j} + \varphi_{0k} \tag{2}$$

where $\gamma_0$ is the intercept term, $\mu_{0j}$ is the cohort effect or residual random effect for cohort $j$, and $\varphi_{0k}$ is the period effect or residual random effect for period $k$.

Combined models:

$$Y_{ijk} = \gamma_0 + \mu_{0j} + \varphi_{0k} + \beta_1 AGE_{ijk} + \beta_2 AGE_{ijk}^2 + \beta_3 X_{ijk} + \beta_4 Z_{ijk} + \varepsilon_{ijk} \tag{3}$$

This model sets the "age" as a fixed effect of individual levels and sets the "period" and "birth queue" into a fine layer group variable above the individual level to avoid the age-period-queue effects Complete common linear relationship.

### 2.2. Data

The data comes from the China Household Tracking Survey (CFPS), which includes data from five publicly available survey points in 2010, 2012, 2014, 2016, and 2018. CFPS is a biennial tracking survey data, aiming to reflect China's economic development and social changes through tracking surveys of a nationally representative sample of villages, households, and household members. In the CFPS tracking survey, the corresponding village data were not publicly available in 2012, 2016, and 2018. Considering that the village data have some stability in the short term, we did the following: the 2012 data were matched with the 2010 village data; the 2016 and 2018 data were matched with the 2014 village data. According to the research needs of this paper, we only selected the group of poor people with agricultural household registration. The identification of poor farmers is mainly based on the "adjusted net household income per capita" indicator in the CFPS household questionnaire, which matches the official income poverty standard lines published by China, namely: RMB 2300 in 2010, RMB 2673 in 2012, RMB 2800 in 2014, RMB 2952 in 2016 and RMB 3200 in 2018. After the statistical processing of the data, the non-poor farmers and the samples with missing values of the main variables were excluded, and finally, 13,559 valid samples were obtained. In the 2010 and 2012 CFPS data, government cash subsidies were subdivided into various subsidies, such as low-income insurance, fallowing, agricultural subsidies, and relief payments. However, starting from the 2014 annual survey data, CFPS no longer provides detailed descriptions of government cash subsidy sources, which are uniformly combined into government subsidies. For the sake of data consistency and comparability, we unified the data of 2010 and 2012 into government subsidies as well.

### 2.3. Variables

The dependent variable is the level of confidence of poor farmers. Lybbert argued that confidences are the desire of individuals based on their observations of those around them [69]. Galiani et al. considered confidence and goals as equivalent concepts [70]. The confidence can be seen as an individual's subjective attitude toward the future realization of self-worth [71,72]. Therefore, we define confidence level as an individual's subjective confidence expectation of the future, which can change according to the individual's conditions and the external environment. In the CFPS database, the "level of confidence in one's future" in the "subjective attitudes" module can adequately reflect the subjective expectations of residents towards the realization of their future values. The variable is divided into one to five levels, from low to high, representing the gradual increase in the level of confidence of rural residents.

The core independent variables are the different poverty reduction policies in the Targeted Poverty Reduction Strategy. Due to the limitations of data accessibility, three types of poverty reduction policies were chosen for this paper: cash grants, transfer employment, and medical insurance.

The temporal dimension variables of interest are age (18 to 70 years), period (2010, 2012, 2014, 2016, and 2018), and birth cohort. The birth cohorts were accordingly divided into 16 cohorts. Births before 1949 and births after 2000 are divided into two separate cohorts, with the remaining years divided into one cohort every three years.

The control variables are divided into basic demographic characteristics variables (gender, ethnicity, health status), social capital and relationship variables (marital status, education level, and subjective socio-economic status), and regional variables (East, Middle, and West). Descriptive statistics are shown in Table 1.

**Table 1.** Descriptive statistics (N = 13,559).

| Variables | Variable Description | Mean | Variance | Min | Max |
|---|---|---|---|---|---|
| Dependent variable | | | | | |
| Confidence | What is your level of confidence in your future? | 3.634 | 1.196 | 1 | 5 |
| Level 1 | | | | | |
| Age | Age of participants | 51.26 | 16.429 | 18 | 70 |
| Gender | Male = 1, Female = 0 | 0.489 | 0.499 | 0 | 1 |
| Ethnic | Han Chinese = 1, Others = 0 | 0.747 | 0.435 | 0 | 1 |
| Health | Yes = 1, No = 0 | 0.721 | 0.448 | 0 | 1 |
| Marriage | Yes = 1, No = 0 | 0.828 | 0.377 | 0 | 1 |
| Education | High school and above = 1, Others = 0 | 0.058 | 0.234 | 0 | 1 |
| Social status | Lower = 1, Higher = 0 | 0.335 | 0.472 | 0 | 1 |
| East | Yes = 1, No = 0 | 0.277 | 0.452 | 0 | 1 |
| Middle | Yes = 1, No = 0 | 0.262 | 0.434 | 0 | 1 |
| West | Yes = 1, No = 0 | 0.461 | 0.498 | 0 | 1 |
| Poverty reduction policies | | | | | |
| Cash grant | Yes = 1, No = 0 | 0.541 | 0.498 | 0 | 1 |
| Transfer employment | Yes = 1, No = 0 | 0.039 | 0.195 | 0 | 1 |
| Medical insurance | Yes = 1, No = 0 | 0.91 | 0.286 | 0 | 1 |
| Level 2 | | | | | |
| Cohort | Birth Cohorts | - | - | 1949 | 2000 |
| period | Survey Years | - | - | 2010 | 2018 |

## 3. Results

### 3.1. Benchmark Regressions

Table 2 shows the baseline regression results of the HAPC-CCREM model, where the first level of fixed effects mainly reports the regression coefficients of pro-poor policies, control variables, and age effects, and the second level of random effects reports the random effects variance and period random effects coefficients. The period effects coefficients are interpreted in a similar way to the general linear model, e.g., a positively significant regression coefficient implies a higher level of confidence than in other survey periods. Model 1 is the base model, which only considers the effects of age, period, and cohort factors on the confidence level of poor farmers; model 2 builds on model 1 by including all individual-level sets of control variables; models 3 to 5 further include different poverty alleviation policies in turn; and model 6 is the full model that introduces all control variables and all poverty alleviation policies. In this section, we first examine the general trend of the change in confidence level of the rural poor in China from 2010 to 2018, and then examine the effect of different poverty alleviation policies on the increase in confidence level and the distribution of this effect across different time dimensions.

**Table 2.** HAPC-CCREM Benchmark Regressions Results (N = 13,559).

| Variables | Model 1 | Model 2 | Model 3 | Model 4 | Model 5 | Model 6 |
|---|---|---|---|---|---|---|
| Fixed effects | | | | | | |
| _cons | 3.683 *** | 3.375 *** | 3.403 *** | 3.397 *** | 3.287 *** | 3.309 *** |
| Age | −0.013 *** | −0.012 *** | −0.012 *** | −0.012 *** | −0.012 *** | −0.012 *** |
| Age Squared | 0.000 ** | 0.000 *** | 0.000 *** | 0.000 *** | 0.000 *** | 0.000 *** |
| Gender | | −0.018 | −0.017 | −0.017 | −0.019 | −0.017 |
| Ethnic | | 0.073 + | 0.074 + | 0.077 + | 0.069 + | 0.073 + |
| Health | | 0.316 *** | 0.314 *** | 0.313 *** | 0.316 *** | 0.313 *** |
| Marriage | | 0.257 *** | 0.255 *** | 0.253 *** | 0.252 *** | 0.249 *** |
| Education | | 0.090 * | 0.088 * | 0.085 * | 0.091 * | 0.086 * |
| Social status | | −0.548 *** | −0.548 *** | −0.548 *** | −0.545 *** | −0.545 *** |
| Reference: Middle | | | | | | |
| East | | 0.02 | 0.014 | 0.013 | 0.021 | 0.014 |
| West | | −0.080 *** | −0.081 *** | −0.080 *** | −0.081 *** | −0.081 *** |
| Poverty reduction policies | | | | | | |
| Cash grant | | | −0.041 * | −0.040 * | | −0.043 * |
| Transfer employment | | | | 0.116 * | | 0.115 * |
| Medical insurance | | | | | 0.099 ** | 0.102 ** |
| Period effects | | | | | | |
| 2010 | −0.275 * | −0.202 + | −0.216 + | −0.215 + | −0.198 + | −0.211 + |
| 2012 | −0.275 * | −0.241 * | −0.236 * | −0.237 * | −0.241 * | −0.237 * |
| 2014 | −0.026 | −0.017 | −0.016 | −0.017 | −0.019 | −0.018 |
| 2016 | 0.182 | 0.123 | 0.13 | 0.129 | 0.12 | 0.126 |
| 2018 | 0.394 ** | 0.337 ** | 0.339 ** | 0.340 ** | 0.338 ** | 0.340 ** |
| Random effects variance | | | | | | |
| Cohort effect | 0.0033 + | 0.001 | 0.001 | 0.001 | 0.0011 | 0.001 |
| Period effect | 0.0854 * | 0.0579 + | 0.0596 + | 0.0595 + | 0.0573 + | 0.0590 + |
| Fitting BIC | 42386.3 | 41326.7 | 41332.5 | 41331.1 | 41327.6 | 41311.5 |

Note: + $p < 0.1$, * $p < 0.05$, ** $p < 0.01$, *** $p < 0.001$. Age and age squared are grand mean-centered.

### 3.2. General Trend Analysis

After controlling for period and cohort effects, both age and age-squared at the individual level had a significant effect on the dependent variable confidence level, and the coefficient on age was consistently negative in all six models, while age-squared positively acted on confidence level, but the coefficient on age-squared was small (less than 0.0001). This shows that there is an approximately linear decreasing relationship between confidence level and age. Figure 1a illustrates this relationship more visually. The confidence level of poor farmers continues to decline with age, and there is a significant negative relationship between age and confidence level. This has been explained from a socio-economic perspective. Young people are energetic, motivated, and free from stress, and therefore have a higher social, psychological, and physical capital, in elders these capitals are to some extent compromised and their level of ambition or confidence decreases [73–75]. Especially in the last stages of life, older people suffer from physical and mental hardships such as illness, widowhood, and retirement, and have a more negative attitude toward the future [76,77]. Some scholars have also argued, based on neuroscientific theories, that when people are young, their brains are most organized and they have a positive desire to explore new and unknown things; as people age, they lose interest in new things because of the loss of brain organization and energy, and become comfortable with the status quo [78,79].

In terms of the period effect, before the implementation of the Targeted Poverty Reduction Strategy, the survey periods of 2010 and 2012 both had a significant negative effect on the confidence level of poor farmers and showed a decreasing trend; while after the formal implementation of the precise poverty alleviation strategy, the trend of the change in the confidence level of poor farmers with the period turned around, and the period effect coefficient of the confidence level of poor farmers in 2018 was significantly positive, Figure 1b visualizes the period effect. It can be seen that since 2013, the confidence

level of the rural poor has shown a monotonic upward trend with the advancement of the year, which to a certain extent reflects that the precise poverty alleviation strategy has a significant poverty reduction effect on the spiritual poverty of rural residents.

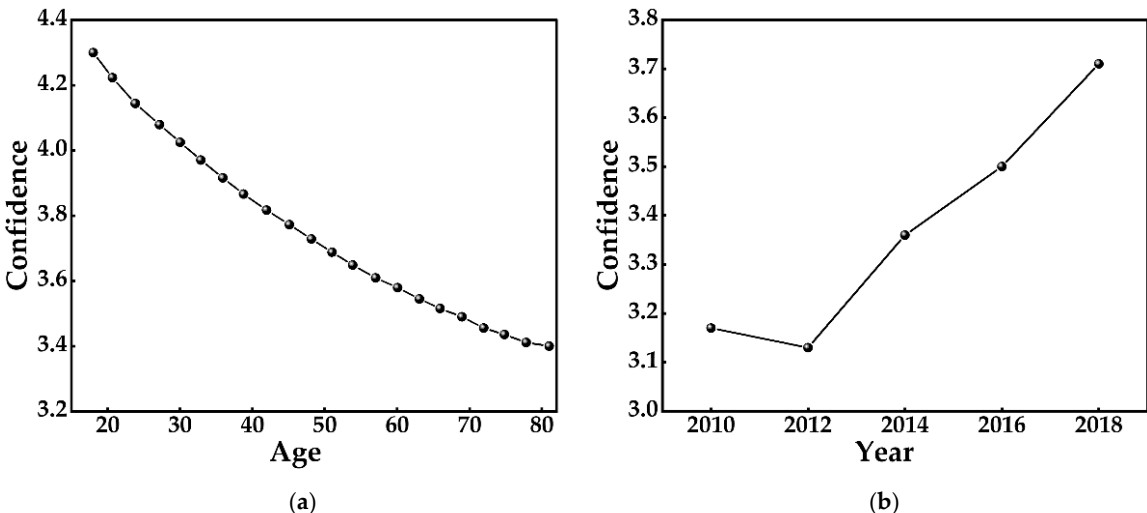

**Figure 1.** Age and period effects of confidence levels of poor farmers. (**a**) is an age effect trend graph, (**b**) is a period effect trend graph.

In terms of random effects variance, the cohort random effect of poor farmers' level of ambition is not significant, i.e., poor farmers in different birth cohorts do not differ significantly in their level of confidence in the future. This has been explained by scholars based on Set Point Theory: confidences or ambitions are more endogenously driven than other values, and each individual is believed to have a specific level of expectation about the future, and events throughout, for life, such as marriage, childbirth, injury, illness, and retirement, may fluctuate to some extent in their ambitions. However, after some time, their level of ambition returns to its initial value. Thus, even if the confidences of poor farmers fluctuate due to specific events, after controlling for period and age effects, their confidences should be evenly distributed at a specific value, so that there is no significant cohort effect.

*3.3. Comparative Analysis of the Impact of Three Poverty Reduction Policies on Confidence*

3.3.1. Cash Grants

Some poor households are not motivated by cash grants and social concern, but instead have developed a "wait-and-see" mentality. Some scholars refer to this phenomenon as the "anti-Hawthorne effect" [80]. The Hawthorne Effect was developed by Mayo, a leading management psychologist, based on the results of the Hawthorne Experiment, in which people prefer to change their behavior in a superior direction when they are aware that they are receiving extra attention [81]. The phenomenon of people choosing not to change or to change their behavior in a less favorable direction when receiving extra attention is known as the "anti-Hawthorne effect". We have found that cash grants, which are poverty alleviation policies, can lead to an 'anti-Hawthorne effect' in the confidence levels of poor farmers.

Figure 2 illustrates this phenomenon visually. In terms of the age effect, before the age of 70, the confidence level of poor farmers who receive government subsidies is significantly lower than that of poor farmers who do not receive government subsidies; it is not until after the age of 70 that the confidence level of poor elderly people with cash subsidies starts to exceed that of poor elderly people without cash subsidies, due to the decline in their physical functions and the reduction in their sources of income. The 'anti-Hawthorne effect' is even more pronounced over time, as the confidence of farmers with cash subsidies is significantly weaker than that of poor farmers without cash subsidies after 2013, and

the gap tends to widen over the years [4]. The reason for the 'anti-Hawthorne effect' can be analyzed in terms of both internal and external factors. Internally, poor farmers who receive cash subsidies tend to be those who have been in deep poverty for a long time, and their starting point for poverty alleviation is generally lower; the common characteristics of this group are a single set of labor skills and a low level of education. As analyzed above, long-term deep poverty may make them more conservative in their mindset and less resilient in escaping poverty [82]. In terms of external factors, the cash subsidy policy focuses more on the mechanical allocation of poverty alleviation resources in the form of "blood transfusion" rigid relief while neglecting the "blood creation" flexible poverty alleviation that focuses on the spiritual needs of poor farmers, which is the main cause of the "anti-Hawthorne effect". This is the main reason for the 'anti-Hawthorne effect', as shown in Figure 3. Of course, due to the limitations of the model, the speculations here have to be further confirmed by subsequent studies.

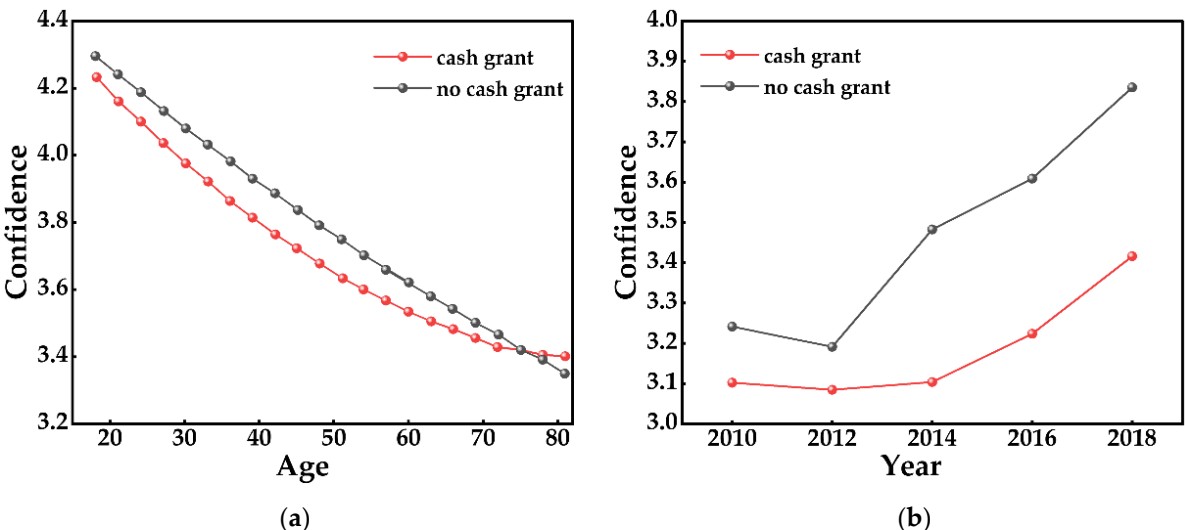

**Figure 2.** The "anti-Hawthorne effect" of cash subsidies on the confidence of farmers. (**a**) is an age effect trend graph, (**b**) is a period effect trend graph.

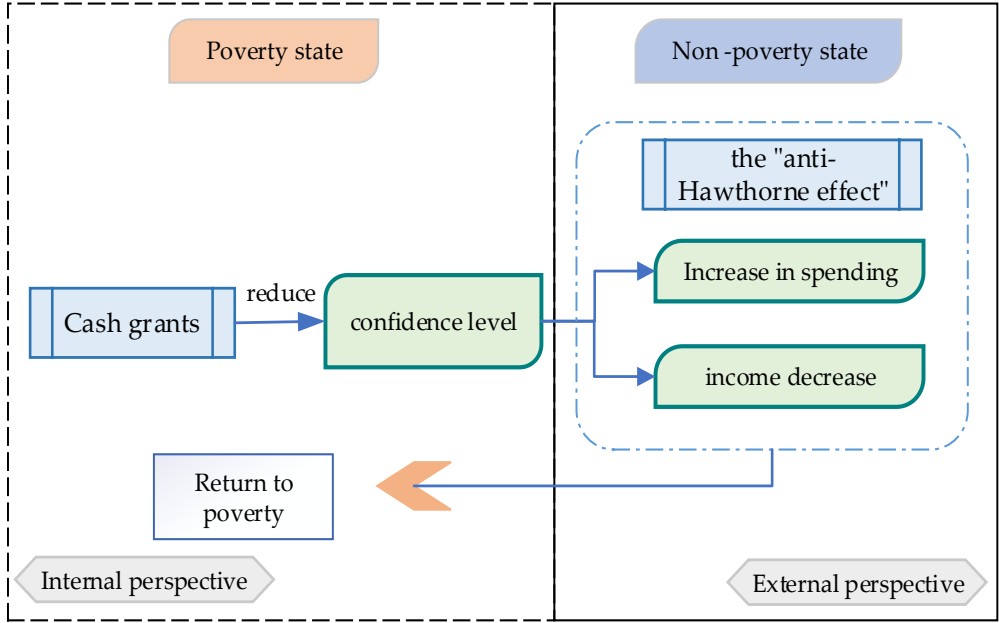

**Figure 3.** Cash grant policy effect mechanism.

TPRS is a process of precisely targeting the basic conditions, causes of poverty and assistance needs of poor groups according to unified standards and through standardised methods and processes. Influenced by single identification indicators and inconsistent standards, as well as emotional tendencies in democratic appraisals, falsification in the establishment of cards and information barriers in data statistics, cases of imprecise identification occur from time to time. The bias created by inaccurate identification is easily exploited by people who take advantage of policy loopholes. Therefore, another explanation for the phenomenon of the poor getting poorer despite the extra funding, is that poor households see being a "poor household" or a "low-income household" as an honor and give up their confidence in surviving on their own efforts under the influence of the psychology of comparison and the dependency of getting something for nothing. Therefore, on 17 November 2018, China's State Council Poverty Alleviation Office and other departments jointly issued the Opinions on Carrying Out Action to Help the Poor and Help the Will, which clearly stipulates that "behaviors such as climbing to follow the trend, not supporting the elderly, and striving to be a poor household should be severely punished and included in the list of defaulters The list of people who have lost trust".

### 3.3.2. Transfer Employment

Figure 4 depicts the trends in the confidence levels of three categories of poor farmers, namely, transfer employment, non-shift employment, and cash subsidy, at different ages and over time. Before the age of 45, the confidence levels of the poor farmers in shift employment are significantly higher than those of the poor farmers in cash subsidy and non-shift employment; between the ages of 45 and 65, the confidence levels of the poor farmers in shift employment tend to be the same; after the age of 65, the confidence levels of the poor farmers in shift employment tend to be the same. Confidence levels of the three groups of poor farmers tend to be the same; while after the age of 65, the confidence level of the shift employment group has increased significantly. If we define the period before the age of 45 as the period of young adulthood, the period between the ages of 45 and 65 is called low old age, and the age above 65 is defined as high old age. This phenomenon can be explained in terms of a life course vision. In young adulthood, working in non-farm jobs or non-farm businesses not only provides an additional source of livelihood income, but also helps individuals bring their physical and psychological energies into play, affirm their self-worth in life, and naturally have a more positive attitude toward life; in the lower old age stage, people's social, psychological, and physical capital is depleted and their desire to work is relatively lower; while the senior old age faces the challenge of old age and infirmity. If they are given a more stable source of income, the effect on their confidence level should be more significant [57].

In terms of period effects, the confidence level of poor farmers who transferred employment was higher than that of poor farmers who did not transfer employment from 2010 to 2018, but the difference between the two showed a convergence trend over the years, which to a certain extent indicates that although transfer employment poverty alleviation has a boosting effect on the confidence level of poor farmers, the effect is not very significant. Comparing the two poverty alleviation policies of transfer employment and cash subsidy, the confidence level of poor farmers who received transfer employment was much higher than that of poor farmers who received cash subsidy in all survey periods. Possible mechanisms of poverty prevention are depicted in Figure 5. This shows that in rural poverty alleviation, the principle of "it's better to teach fishing than to offer fish" should be adopted, effectively improving farmers' livelihood skills.

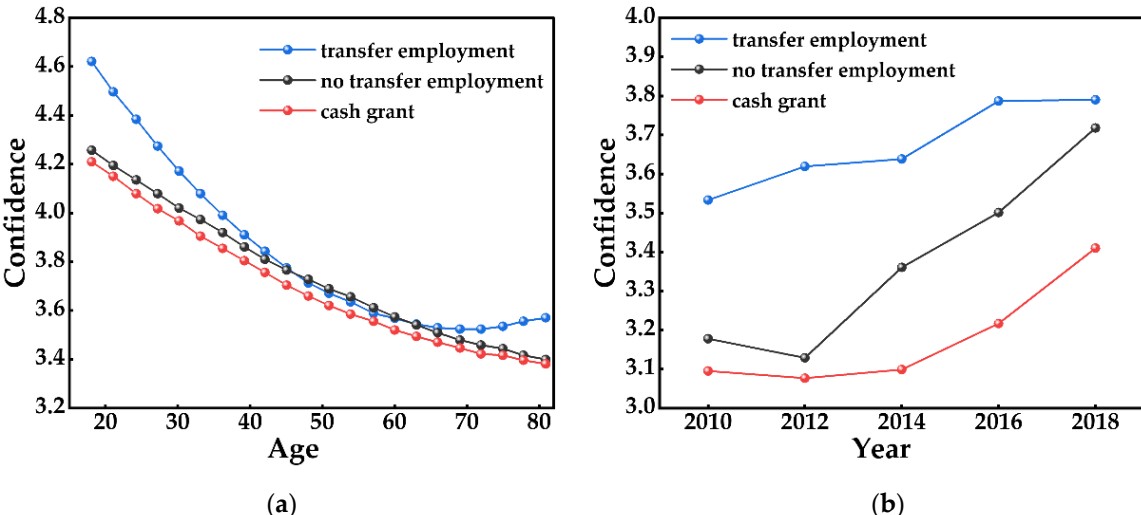

**Figure 4.** Age and period effects of transfer employment on confidence. (**a**) is an age effect trend graph, (**b**) is a period effect trend graph.

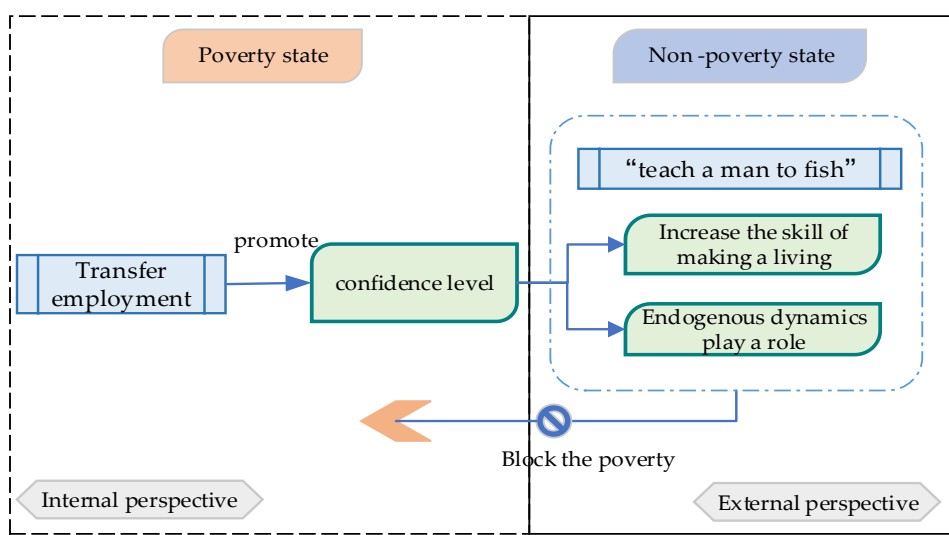

**Figure 5.** Transfer employment policy effect mechanism.

### 3.3.3. Medical Insurance

As shown in Figure 6a, before the age of 40, the effect of health insurance on the confidence level of poor farmers is minimal, with the age trend lines of the two almost coinciding, but after the age of 40, the confidence level of poor farmers with health insurance is significantly higher than that of poor farmers without health insurance, and the gap tends to widen as age increases. This indicates that medical insurance can have a significant effect on the confidence level of the rural poor at an advanced age. Poverty due to illness is one of the most important causes of poverty in rural China, especially among the elderly and vulnerable groups, whose ability to withstand the risk of illness is particularly weak. On the one hand, health insurance has contributed to a change in farmers' attitudes toward the risk of illness, with over 60% of farmers feeling psychologically protected against the risk of illness when they have health insurance. On the other hand, by participating in health insurance, poor farmers can seek medical treatment promptly, preventing minor illnesses from becoming serious ones, reducing their medical and financial expenses, and improving their health.

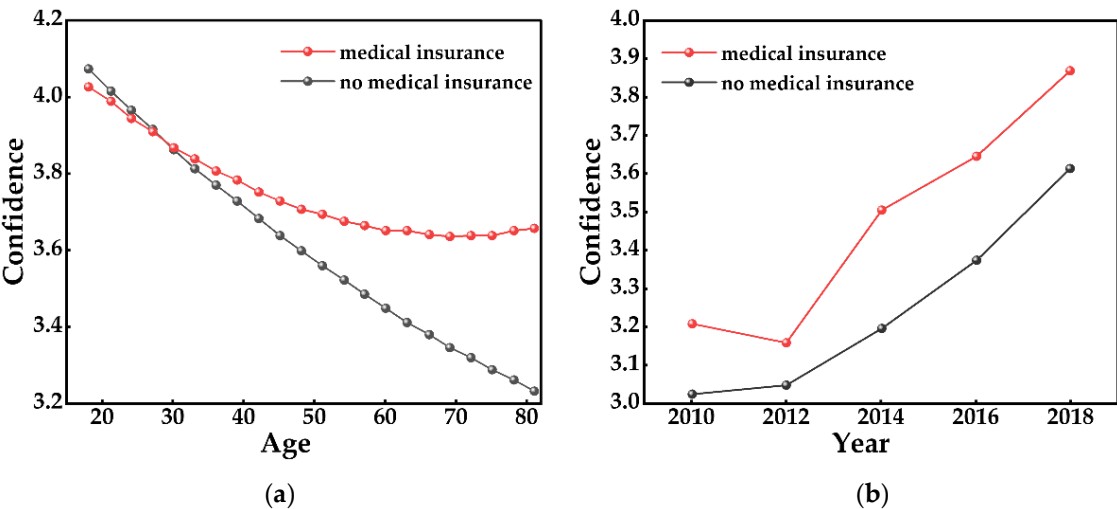

**Figure 6.** Age and period effects of medical insurance on confidence. (**a**) is an age effect trend graph, (**b**) is a period effect trend graph.

As shown in Figure 6b, between 2010 and 2018, the confidence levels of poor farmers with medical insurance were significantly higher than those of poor farmers without medical insurance, and, after 2013, the gap in confidence levels between the two tended to widen. On the one hand, China has implemented a precise poverty alleviation strategy, increased the reimbursement ratio and coverage of rural medical insurance, and gradually included chronic diseases such as hypertension and diabetes in the coverage. On the other hand, the accelerating aging of China's rural areas and the rapid rise in the proportion of elderly people may also have contributed to a significant divergence in the confidence levels of poor farmers.

## 4. Discussion

Effectively stimulating the aspiration level of poor households is the key to forming an endogenous motivation for self-development, which is the consensus of studies related to precise poverty alleviation policies; however, existing studies tend to examine poverty at a point in time-based on macroscopic cross-sectional data, ignoring the impact of the broader context of social change on self-confidence levels and the heterogeneity of poverty reduction strategies in different age dimensions. At the same time, one-dimensional or static studies are likely to obscure valuable findings or even draw conflicting or completely factually incorrect conclusions. The APC approach adopted in this paper precisely complements the above research deficiencies by focusing on the effects of era context and age distribution on the correlations.

This paper analyzes the heterogeneous relationship between confidence levels of different age groups under different historical times and influenced by different poverty alleviation policies with the perspective of endogeneity of poverty alleviation. The results find that the relationship between different poverty alleviation policies and poor farmers' confidence is not static or linearly changing in the process of social change but has a historical and group heterogeneous change process. This paper divides the precise poverty alleviation policies into financial subsidies, employment transfer, and medical protection, and finds the corresponding "anti-Hawthorne effect", "teaching to fish", and "bottom-up effect" three effects, i.e., different paths through which farmers' self-confidence is inhibited or stimulated to different degrees. This interesting finding complements the research on the endogenous dynamics of poverty alleviation policies and provides theoretical guidance for the government to evaluate the effects of different policy tools on mental poverty alleviation.

Although this paper does analyze the different effects of different policies, historical backgrounds, and age groups on poverty alleviation policies on the level of self-confidence, it does so with some caveats. For instance, important historical events in different historical

backgrounds, and stages of development need to be further explored and analyzed. In addition, due to the lack of data, no further quantitative analysis was done on the level of impact of different poverty alleviation policies on the level of self-confidence. In the future, we will conduct in depth research on the aforementioned topics to arrive at more meaningful conclusions and offer more instructive recommendations for practice.

## 5. Conclusions

Confidence is a key internal factor in lifting rural residents out of poverty, and a high level of confidence can stimulate people's endogenous drive for development. We utilized APC to explore the pure effects of poverty reduction policies on farmers' confidence to inform rural poverty reduction policies within TPRS. The following are our findings:

(1) Cash grants have an "anti-Hawthorne effect", farmers who receive cash grants have significantly lower levels of confidence than those who do not receive cash grants. This phenomenon is more obvious among people aged 18–70 and the gap continued to widen between 2013 and 2018. The risk of returning to poverty from the internal perspective mainly stems from the unsustainability of the drivers of poverty eradication, which was mainly due to the external help being mostly material, which led to a dependency mentality. The risk of returning to poverty from an external perspective is mainly due to the reduction in the living standard of households that have been lifted out of poverty due to the lack of assets and the inability to play the role of a "protection net" of assets. The risk of returning to poverty is very high once the financial assistance stops.

(2) "It' s better to teach a man fishing than to give him fish". Transfer employment is more likely to upgrade farmers' confidence level than cash grants, and this gap is most pronounced among young adults (18–45 years) and older people (65+ years). The motivation to escape from poverty from the internal perspective mainly originates from endogenous motivation, which forms confidence from support to autonomy through help. From the external perspective, the farmers being helped improve the objective ability of the poor target to escape from poverty due to the diversification of the means of help initiatives implemented by the outside world. Through the link of self-confidence and the precipitation of time, a sustainable state of improvement can be formed, fundamentally blocking the risk of returning to poverty.

(3) Medical insurance policy plays an important role in reducing worries. The confidence of farmers with medical insurance is much higher than that of farmers without medical insurance, and this gap has continued to expand with age and years. From an internal perspective, the security measures create inner stability and strengthen the foundation for upward development. From the external perspective, health insurance plays the role of a "protection cushion", and self-confidence is improved to a certain extent, which has the effect of stopping the return to poverty to a certain extent.

Lessons for TPRS suggest that poverty reduction strategies should incorporate psychological variables into the effectiveness evaluation of poverty alleviation projects, focus on the self-confidence-boosting effects of employment, and pay attention to the mental poverty of rural elderly people to achieve sustainable and stable poverty reduction among poor groups. Based on the above research results, the following relevant policy recommendations are proposed.

(1) Include psychological variables in the performance appraisal of poverty reduction strategies. In the evaluation and assessment process, the government should include the psychological benefits of poverty reduction strategies and the negative psychological consequences caused by poverty in the comprehensive assessment of projects and introduce psychological variables into poverty reduction to achieve the targeted objectives of the intervention. The long-term psychological impact on poor farmers should be dynamically evaluated, focusing not only on the satisfaction of poor farmers with the poverty reduction policy but also on the psychological changes of poor residents before and after receiving policy support to improve the overall effectiveness of poverty reduction.

(2) Focus on the aspirational failures of the poor rural elderly. The poor rural elderly have the lowest level of confidence and the most significant effect of pro-poor policies on raising their level of confidence. This may be related to their physical functioning, psychological state, and changes in the retirement environment. Therefore, in the context of aging, on the one hand, it is important to encourage and guide the financial management behavior and pension savings of the elderly in rural areas, to alleviate the financial pressure in their old age; on the other hand, it is important to innovate the pension model, instead of simply treating all the elderly in rural areas as a pension group or a group of people to be supported, to introduce the elderly workforce into the pension service system by transferring employment. For example, exploring and developing a mutual help pension model for the elderly to create new labor opportunities for the elderly in rural areas while at the same time providing them with a pension option that is relatively easy on their financial burden.

(3) Pay attention to the poverty-reducing effects of employment. As the old saying goes, "It' s better to teach a man fishing than to give him fish." Poor farmers who receive cash subsidies are likely to have lower levels of ambition, while employment can significantly promote poor farmers' confidence levels. This suggests that poverty reduction strategies need to shift from "blood transfusion" rigid relief to "blood creation" flexible poverty alleviation. Poverty reduction strategies at this stage are often devoted to addressing the external causes of poverty while neglecting the psychological burden of the poor caused by poverty. To achieve poverty eradication, it is necessary to adopt a poverty alleviation policy that takes into account both internal and external factors, inspire confidence in poor residents themselves to achieve poverty eradication, and guards against confidences failure to ensure the achievement of sustainable and stable poverty eradication for the poor groups. The government should play the role of guiding the market, helping poor residents to diversify and share market and natural risks, and reducing the poor group's negative expectations of production and life. We should also give full play to the role of government guidance and market mechanisms to help poor residents spread and share market risks and natural risks, to reduce their negative expectations of production and living goals and raise their level of confidence to get out of poverty and become rich.

**Author Contributions:** Conceptualization, Z.W.; methodology, M.Y.; software, M.Y. and Y.S.; validation, Z.W. and K.G.; formal analysis, Z.W., M.Y., K.G. and Y.S.; investigation, Z.W.; resources, K.G.; data curation, M.Y.; writing—original draft preparation, Z.W. and M.Y.; writing—review and editing, M.Y., K.G. and Z.Z.; visualization, Z.W.; supervision, M.Y. All authors have read and agreed to the published version of the manuscript.

**Funding:** This research was financially supported by the National Social Science Foundation of China (Grant No. 20&ZD095), Pan Jiahua Expert Workstation (Grant No. 2021GZZH01), the Youth Program of the National Social Science Foundation of China (grant 21CJY044), a key program of Guizhou Philosophy and Social Sciences Planning (grant 21GZD60), the Philosophy and Social Sciences Planning Project of Shanxi Province (No. 2021YJ078).

**Institutional Review Board Statement:** Not applicable.

**Informed Consent Statement:** Not applicable.

**Data Availability Statement:** Not applicable.

**Acknowledgments:** The authors thank the China Family Panel Studies (CFPS) for providing the data.

**Conflicts of Interest:** The authors declare no conflict of interest.

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
