# Peer review of "Evolution in the Impact of Pro-Poor Policies on Farmers’ Confidence: Based on Age-Period-Cohort Analysis Perspective"

_sustainability, doi:10.3390/su151310525_

Round 1

Reviewer 1 Report

This is an interesting and novel of piece of research. The use of the age-cohort and period interactions is suitable and the topic is very important. 

However, the authors make strong conclusions throughout the manuscript without sufficient supporting evidence. In the introduction, the authors refer to an article by Dalton et al. published in the Economic Journal. The conclusion of that particular article is much more nuanced than is described in this manuscript.

The manuscript is mainly well-written. However, there are several instances when the writing departs from an academic style e.g. 'Confidence is more important than gold' (Line 41-42).

The authors should refrain from referring to poor farmers as 'they' e.g. line 84. In the introduction, a reference is made to an article by Goldstein et al. However, it is not clear  from the text as to the relevance of that article for this research.

In the data section, the authors need to provide more background and explanation for the cash grants variables including the eligibility criteria and the rate of take-up. The income variables needs more explanation. It appears from the text that the income variable is adjusted but it is not clear if equivalence scales are applied. The social status variable needs more explanation that is currently reported.

The interpretation of the results in Figure 1a could be improved. The authors place a lot of emphasis on the so-called 'anti-Hawthorne effect', which appears excessive given the lack a specific variable about this effect.

In the results section, Table 2 appears good. However, it is recommended that the constant does not appear on the first row.

Figure 3 needs improvement as it is not very clear. The authors find a positive relationship between medical insurance and confidence levels. However, the vast majority of cases avail of medical insurance. There is a need to explain the reasons for the non-adoption of medical insurance for the small minority (9 per cent) who do not adopt medical insurance.

In the first paragraph of the conclusion section, the authors need to be upfront as to whether or not their methods are newly developed by the authors or based on previous methodological innovations by previous authors (Line 414). The conclusions appear overconfident about the causal effects and the authors need to acknowledge the potential for omitted variable bias.

The references could include the following:

https://www.mdpi.com/2077-0472/11/11/1075

The manuscript is mainly well-written. However, there are several instances when the writing departs from an academic style e.g. 'Confidence is more important than gold' (Line 41-42). The authors should refrain from referring to poor farmers as 'they' e.g. line 84. The authors should consider alternatives to 'blood transfusion' and 'blood creation'. Please ensure that there is spacing after each full stop.

Author Response

Response to Reviewer Comments

Overall Response: We thank the editor and the anonymous referees very much for constructive and detailed comments. We have seriously considered and addressed all the comments in the revised manuscript. Our point-to-point responses are provided follows. Also, to facilitate the review process, the major changes are highlighted in red in the revised manuscript.

Point 1: The manuscript is mainly well-written. However, there are several instances when the writing departs from an academic style e.g. 'Confidence is more important than gold' (Line 41-42).

Response 1: Thanks for your suggestions. The study by Dalton et al. published in the Economic Journal demonstrates that the root cause of chronic poverty is a lack of self-confidence, and therefore, regaining confidence is also key to the success of pro-poor policies, which is the underlying logic of this study.

Point 2: The manuscript is mainly well-written. However, there are several instances when the writing departs from an academic style e.g. 'Confidence is more important than gold' (Line 41-42).

Response 2: Thanks for your suggestions. Based on your suggestion, we have removed the content of "Based on your suggestion, we have removed the content of ".

Point 3: The authors should refrain from referring to poor farmers as 'they' e.g. line 84. In the introduction, a reference is made to an article by Goldstein et al. However, it is not clear  from the text as to the relevance of that article for this research.

Response 3: Thanks for your suggestions. Per your suggestion, We have rewritten the description. After consideration, we decided to delete the third paragraph of the preamble as it was not very relevant to the topic.

Point 4: In the data section, the authors need to provide more background and explanation for the cash grants variables including the eligibility criteria and the rate of take-up. The income variables needs more explanation. It appears from the text that the income variable is adjusted but it is not clear if equivalence scales are applied. The social status variable needs more explanation that is currently reported.

Response 4: Thanks for your suggestions. This is important. The conditions for obtaining funding are fixed in the same period and have no effect on the conclusions of this study. In the APC model, the effect of the APC model refers to the trend of self -confidence influenced by different poverty alleviation policies in different periods. At the same time, because the CFPS database is adopted in this article, there is no detailed explanation of policy funding conditions, and it cannot provide a description.

Point 5: The interpretation of the results in Figure 1a could be improved. The authors place a lot of emphasis on the so-called 'anti-Hawthorne effect', which appears excessive given the lack a specific variable about this effect.

Response 5: Thanks for your suggestions. In the "3.2.1.cash Grants" section, 410-419, the source and significance of the "Anti-Hawthorne Effect" was introduced.

Point 6: In the results section, Table 2 appears good. However, it is recommended that the constant does not appear on the first row.

Response 6: Thanks for your suggestions. We have changed the format of Table 2 in the revised version.

Point 7: Figure 3 needs improvement as it is not very clear. The authors find a positive relationship between medical insurance and confidence levels. However, the vast majority of cases avail of medical insurance. There is a need to explain the reasons for the non-adoption of medical insurance for the small minority (9 per cent) who do not adopt medical insurance.

Response 7: Thanks for your suggestions. Figure 3 is made with VISO software. We replaced it with clearer pictures. We have paid attention to this question you mentioned. Chinese medical insurance is voluntarily participated. Some farmers do not participate in medical insurance. The following three points: 1. Farmers believe that the current payment is very high. It is considered to be a waste of this money; 3. Medical insurance is mainly a guarantee system for the reimbursement of hospitalization for farmers after the birth of a serious illness. The illusion of use. These reasons may not have a great impact on this article. In the future research, we will pay attention to this issue.

Point 8In the first paragraph of the conclusion section, the authors need to be upfront as to whether or not their methods are newly developed by the authors or based on previous methodological innovations by previous authors (Line 414). The conclusions appear overconfident about the causal effects and the authors need to acknowledge the potential for omitted variable bias.

Response 8: Thanks for the comment.

Point 9The references could include the following:https://www.mdpi.com/2077-0472/11/11/1075

Response 9: Thanks for the comment. We have carefully learned the article you recommend, which is greatly inspired by this study and cited it in the article. Quote for reference 34.

Point 10The manuscript is mainly well-written. However, there are several instances when the writing departs from an academic style e.g. 'Confidence is more important than gold' (Line 41-42). The authors should refrain from referring to poor farmers as 'they' e.g. line 84. The authors should consider alternatives to 'blood transfusion' and 'blood creation'. Please ensure that there is spacing after each full stop.

Response 10: Thanks for the comment. In response to the problem of language expression, we have made corrections one by one, thank you for the question raised

In closing, we thank you again for helping us to improve the quality of this paper.

Reviewer 2 Report

Review report attached.

Minor editing required for grammatical errors. 

Author Response

Response to Reviewer Comments

Overall Response: We thank the editor and the anonymous referees very much for constructive and detailed comments. We have seriously considered and addressed all the comments in the revised manuscript. Our point-to-point responses are provided follows. Also, to facilitate the review process, the major changes are highlighted in red in the revised manuscript.

Point 1: The first part of the introduction is well-written. But authors say that they are trying to measure mental/psychological poverty, what they do is only the confidence offarmers in their future.that too not defined in what respect. Authors need to make sure that they are measuring psychological poverty or not. For example, questions like “Do you think you will have more food/more income or better employment opportunities?” will better measure a mental/psychological poverty aspect.

Response 1: Thanks for your suggestions. The level of confidence in this study comes from the CFPS database. The questionnaire question is: Do you have the degree of confidence in your future? From 1 to 5 points into 5 levels.

Point 2: In the paper, authors included people from the age group of 18-97. Why is it needed to include people beyond the age of 65+, Off-course people do not see much of their future (especially in employment) after the age if 65.

Response 2: Thanks for your suggestions. Based on the sampling time of this research data, after we eliminated the people born before and after 2000, we were divided into 16 groups according to the age, and divided into 16 groups. The processing method explains the 328-331 of "2.2. Variables".

Point 3: I also wonder if there are some eligibility criteria to be selected for the government transfers (in employment transfer specially).

Response 3: Thanks for your suggestions. Employment transfer refers to the process of not being engaged in agricultural production activities, or insufficient agricultural labor time for various reasons. It is necessary to work and do business in industrial and service industries other than agricultural production to obtain legal income. Any rural labor that meets the requirements of transfer employment can go through the employment registration procedures and include the unified employment management service system in the city. The employment transfer data designed in this study comes from the CFPS database question volume investigation data.

Point 4: Authors say that poverty is defined based on net household income per capita. How does this correspond to the international measure?

Response 4: Thanks for your suggestions. The identification of poor farmers is mainly based on the "adjusted net household in-come per capita" indicator in the CFPS household questionnaire, which matches the official income poverty standard lines published by China, namely: RMB 2,300 in 2010, RMB 2,673 in 2012, RMB 2,800 in 2014, RMB 2,952 in 2016 and RMB 3,200 in 2018.

Point 5: The definition of the variables used in the analysis needs to be clearly explained. How is health variable defined? ls it self-rated health? On what scale? What does"1 and “0" represent? How is social status categorized in “Lower” and “Higher”? What is the time framework for poverty reduction policies? During what time do they receive cash grant/transfer employment ormedical insurance?.

Response 5: Thanks for your suggestions. The APC method used in this article focuses on the trend of age effects and period effects. This research data comes from the public database CFPS. The health indicators come from the authentic judgment and answer. Whether or not the questionnaire was determined according to the survey data of the year.

Point 6: There is also a huge endogeneity issue. For example, people who are poor and so have low6confidence, they are majorly the people who receive government transfers.

Response 6: Thanks for your suggestions. This study compares the samples according to different groups of policies. The APC method adopted shows the changes in the trend of time and period effects. There is no endogenous problem.

Point 7: Another observation from methodology. When authors used fixed effects, they should also beable to report the coefficients for the time varying variables like age, health or the povertyreduction variables (if that changed) because individual level factors that are fixed over timesuch as Gender, ethnicity, regions, cohorts should be fixed.

Response 7: Thanks for your suggestions. This study only pays attention to the changes in the level of confidence in age and period. The CFPS database is to track and survey data. The characteristics of individual races, gender, region and other characteristics are fixed.

In closing, we thank you again for helping us to improve the quality of this paper.

Round 2

Reviewer 1 Report

The eligibility and adoption decisions in relation to cash grants needs much more detail.

Endogeneity is not discussed sufficiently in the revised manuscript (exception page 12 about the age-cohort method). This is the reason for the 'must improved' recommendations. Endogeneity needs to be discussed in more detail (potential omitted variable bias or reverse causality) and how the use of the fixed effects model may correct for this potential bias.

Satisfactory

Author Response

Response to Reviewer Comments

Overall Response: We thank the editor and the anonymous referees very much for constructive and detailed comments. We have seriously considered and addressed all the comments in the revised manuscript. Our point-to-point responses are provided follows.

Point 1: The eligibility and adoption decisions in relation to cash grants needs much more detail.

Response 1: Thanks for your suggestions. In the 2010 and 2012 CFPS data, government cash subsidies were subdivided into various subsidies, such as low income insurance, fallowing, agricultural subsidies, and relief payments. However, starting from the 2014 annual survey data, CFPS no longer provides detailed descriptions of government cash subsidy sources, which are uniformly combined into government subsidies. For the sake of data consistency and comparability, we unified the data of 2010 and 2012 into government subsidies as well. In the revised version we have added source notes to the table, please refer to lines 218-224.

Point 2: Endogeneity is not discussed sufficiently in the revised manuscript (exception page 12 about the age-cohort method). This is the reason for the 'must improved' recommendations. Endogeneity needs to be discussed in more detail (potential omitted variable bias or reverse causality) and how the use of the fixed effects model may correct for this potential bias.

Response 2: Thanks for your suggestions. Your suggestion is very useful, and We have sorted out the APC methods used and the logic of the arguments, The explanation is as follows.

The APC model allows for simple regression analysis of microsample data, Yang noted that this may violate the independence-of-errors assumption on which conventional fixed-effects regression models (e.g., ordinary least squares or logit regression) are based. They developed a hierarchical age-period-cohort models (HAPC models) approach to address this problem. Specifically, they applied cross-classified random-effects two-level models (CCREM) to repeated survey data to ascertain whether there are any clustering effects in survey responses by higher level units—namely, the survey time period and birth cohort. Yang (forthcoming-b) studied and compared the performance of restricted maximum likelihood (REML), empirical Bayes (EB), and full Bayes (FB) estimators in the context of this HAPC approach to micro APC data. The methodology of this study focused on learning the HAPC model of Yang, Y. See references 52, 55, 56, 57, 58 for details. The HAPC model in the references of this paper is not tested for endogeneity, and this study is more concerned with the change of trend under age and period.

TPRS is a process of precisely targeting the basic conditions, causes of poverty and assistance needs of poor groups according to unified standards and through standardised methods and processes. Influenced by single identification indicators and inconsistent standards, as well as emotional tendencies in democratic appraisals, falsification in the establishment of cards and information barriers in data statistics, cases of imprecise iden-tification occur from time to time. The bias created by inaccurate identification is easily exploited by people who take advantage of policy loopholes. Therefore, another explana-tion for the phenomenon of the poor getting poorer the more funding is that poor house-holds see being a "poor household" or a "low-income household" as an honour and give up their confidence in surviving on their own efforts under the influence of the psychology of comparison and the dependency of getting something for nothing. Therefore, on No-vember 17, 2018, China's State Council Poverty Alleviation Office and other departments jointly issued the Opinions on Carrying Out Action to Help the Poor and Help the Will, which clearly stipulates that "behaviors such as climbing to follow the trend, not support-ing the elderly, and striving to be a poor household should be severely punished and in-cluded in the list of defaulters The list of people who have lost trust". See lines 382-397 in the revised version.

In closing, we thank you again for helping us to improve the quality of this paper.

Reviewer 2 Report

NA

NA

Author Response

Response to Reviewer Comments

Overall Response: We thank the editor and the anonymous referees very much for constructive and detailed comments. We have seriously considered and addressed all the comments in the revised manuscript.

Point 1: Additional explanation of data sources and processing.

Response 1: In the 2010 and 2012 CFPS data, government cash subsidies were subdivided into various subsidies, such as low income insurance, fallowing, agricultural subsidies, and relief payments. However, starting from the 2014 annual survey data, CFPS no longer provides detailed descriptions of government cash subsidy sources, which are uniformly combined into government subsidies. For the sake of data consistency and comparability, we unified the data of 2010 and 2012 into government subsidies as well. In the revised version we have added source notes to the table, please refer to lines 218-224.

Point 2: The APC approach to endogeneity used in this study was sorted out.

Response 2: The APC model allows for simple regression analysis of microsample data, Yang noted that this may violate the independence-of-errors assumption on which conventional fixed-effects regression models (e.g., ordinary least squares or logit regression) are based. They developed a hierarchical age-period-cohort models (HAPC models) approach to address this problem. Specifically, they applied cross-classified random-effects two-level models (CCREM) to repeated survey data to ascertain whether there are any clustering effects in survey responses by higher level units—namely, the survey time period and birth cohort. Yang (forthcoming-b) studied and compared the performance of restricted maximum likelihood (REML), empirical Bayes (EB), and full Bayes (FB) estimators in the context of this HAPC approach to micro APC data. The methodology of this study focused on learning the HAPC model of Yang, Y. See references 52, 55, 56, 57, 58 for details. The HAPC model in the references of this paper is not tested for endogeneity, and this study is more concerned with the change of trend under age and period.

Point 3Explains the lower confidence under cash grants.

Response 3: TPRS is a process of precisely targeting the basic conditions, causes of poverty and assistance needs of poor groups according to unified standards and through standardised methods and processes. Influenced by single identification indicators and inconsistent standards, as well as emotional tendencies in democratic appraisals, falsification in the establishment of cards and information barriers in data statistics, cases of imprecise iden-tification occur from time to time. The bias created by inaccurate identification is easily exploited by people who take advantage of policy loopholes. Therefore, another explana-tion for the phenomenon of the poor getting poorer the more funding is that poor house-holds see being a "poor household" or a "low-income household" as an honour and give up their confidence in surviving on their own efforts under the influence of the psychology of comparison and the dependency of getting something for nothing. Therefore, on No-vember 17, 2018, China's State Council Poverty Alleviation Office and other departments jointly issued the Opinions on Carrying Out Action to Help the Poor and Help the Will, which clearly stipulates that "behaviors such as climbing to follow the trend, not support-ing the elderly, and striving to be a poor household should be severely punished and in-cluded in the list of defaulters The list of people who have lost trust". See lines 382-397 in the revised version.

In addition we have made changes to English language.

In closing, we thank you again for helping us to improve the quality of this paper.
